# One-Year Outcomes after Ledipasvir/Sofosbuvir Treatment of Chronic Hepatitis C in Teenagers with and without Significant Liver Fibrosis—A Case Series Report

**DOI:** 10.3390/v13081518

**Published:** 2021-07-31

**Authors:** Maria Pokorska-Śpiewak, Anna Dobrzeniecka, Magdalena Marczyńska

**Affiliations:** 1Department of Children’s Infectious Diseases, Medical University of Warsaw, 01-201 Warsaw, Poland; magdalena.marczynska@wum.edu.pl; 2Department of Pediatric Infectious Diseases, Regional Hospital of Infectious Diseases in Warsaw, 01-201 Warsaw, Poland; adobrzeniecka@zakazny.pl

**Keywords:** children, cirrhosis, direct acting antiviral, hepatitis C virus, liver fibrosis

## Abstract

One-year outcomes after therapy with ledipasvir/sofosbuvir (LDV/SOF) in children with chronic hepatitis C (CHC) presenting with and without significant liver fibrosis were analyzed. We included patients aged 12–17 years treated with LDV/SOF, presenting with significant fibrosis (F ≥ 2 on the METAVIR scale) in transient elastography (TE) at the baseline and we compared the outcomes with that of patients without fibrosis. Patients were followed every 4 weeks during the treatment, at the end of the therapy, at week 12 posttreatment, and one year after the end of treatment. Liver fibrosis was established using noninvasive methods: TE, aspartate transaminase-to-platelet ratio index (APRI), and Fibrosis-4 index (FIB-4). There were four patients with significant fibrosis at baseline: one with a fibrosis score of F2 on the METAVIR scale, and three with cirrhosis (F4) at baseline. One year after the end of treatment, the hepatitis C viral load was undetectable in three of them. One patient was lost to follow-up after week 4. In two out of the four patients, a significant improvement and regression of liver fibrosis was observed (from stage F4 and F2 to F0-F1 on the METAVIR scale). In one patient, the liver stiffness measurement median increased 12 weeks after the end of the treatment and then decreased, but still correlated with stage F4. An improvement in the APRI was observed in all patients. In four patients without fibrosis, the treatment was effective and no progression of fibrosis was observed. A one-year observation of teenagers with CHC and significant fibrosis treated with LDV/SOF revealed that regression of liver fibrosis is possible, but not certain. Further observations in larger groups of patients are necessary to find predictors of liver fibrosis regression.

## 1. Introduction

In the majority of cases, hepatitis C virus (HCV) infection in children leads to chronic hepatitis C (CHC) with a possible progression to serious liver disease. The risk of cirrhosis in HCV-infected children and adolescents was estimated at 1 to 2% [1,2,3,4,5]. However, a number of recent observational studies have demonstrated that a higher proportion of pediatric patients than had been previously thought may develop advanced liver disease resulting from HCV infection [6,7,8]. Turkova et al. demonstrated bridging fibrosis in 41% of their patients by histopathological evaluation with a median age of 10.4 years, and liver stiffness measurement (LSM) over 5.0 kPa in transient elastography (TE) evaluation in 30% of patients [7]. Modin et al. analyzed the outcome of CHC in 1049 patients infected with HCV during childhood and found that serious liver disease developed in 32% of cases with a median time of 33 years after infection, irrespective of the mode of infection [8]. Thus, children infected vertically developed cirrhosis at an earlier age compared to patients infected from other sources, which is consistent with other observations [4]. The risk of HCV-related hepatocellular carcinoma (HCC) was 5%, the incidence of liver transplant was 4%, and the risk of death was 3% [8]. New highly effective interferon-free therapies for HCV infection based on direct-acting antivirals (DAAs) may be helpful for the prevention of long-term liver disease progression related to HCV by eliminating the virus [8,9]. In a recent systematic review with meta-analysis on the efficacy and safety of DAAs in children and adolescents, Indolfi et al. demonstrated that among the patients receiving all doses of treatment, 100% of cases reached a sustained virologic response (SVR) [9]. Among the subjects receiving at least one dose of DAA, the lowest efficacy rates were observed among cirrhotic patients (83%) [9]. However, only a few outcomes of DAA treatment in cirrhotic pediatric patients have been reported, and there is scarce and limited data on the influence of these therapies on liver fibrosis [9,10]. We recently reported that among our 35 patients aged 12–17 years qualifying for ledipasvir/sofosbuvir (LDV/SOF) treatment, 11% of cases presented with significant fibrosis (F ≥ 2 on the METAVIR scale), including 9% with cirrhosis [6]. Thus, in this paper, we aim to present the one-year outcomes after therapy in this specific group of patients. In particular, the influence of LDV/SOF treatment on the extent of liver fibrosis was analyzed.

## 2. Materials and Methods

### 2.1. Study Group

In our single tertiary health care pediatric infectious disease center, consecutive HCV-infected patients aged 12–17 years, treated between August 2019 and February 2020, were qualified for the real-life therapeutic program ‘Treatment of Polish Adolescents with Chronic Hepatitis C Using Direct Acting Antivirals (*POLAC* PROJECT)’. Patients infected with genotypes 1 and 4 of the HCV were treated with sofosbuvir/ledipasvir (SOF/LDV), which was available courtesy of a donation by a pharmaceutical company. Patients with CHC (diagnosed in subjects with over a 6 month duration of disease confirmed with positive nucleic acid testing, HCV RNA, using a quantitative real-time polymerase chain reaction, RT-PCR, Abbott Real Time HCV, Abbott Laboratories, Abbott Park, IL, USA; measurement linearity range 12–1.0 × 10^8^ IU/mL) were qualified for treatment irrespective of the extent of liver fibrosis or previous ineffective treatment. The duration of treatment was established according to the recommendations of the European Society of Paediatric Gastroenterology, Hepatology and Nutrition (ESPGHAN): patients received 12 weeks of therapy unless they were infected with HCV genotype 1 with a history of previous ineffective interferon-based treatment and presented with cirrhosis [11]. Patients in this study were followed every 4 weeks during the treatment, at the end of the therapy, at week 12 posttreatment (to assess the efficacy of the treatment based on a sustained virologic response, SVR12), and one year (52 weeks) after the end of treatment to assess long-term outcomes. Body mass index standard deviation (SD) scores (BMI z-scores) were calculated according to the WHO Child Growth Standards and Growth reference data using the WHO Anthropometric calculator AnthroPlus v1.0.4 (World Health Organization, Geneva, Switzerland). Obesity was diagnosed in children with a BMI z-score > 2 SD and overweight children with a BMI z-score > 1 SD. 

In this prospective study, we analyzed a group of patients presenting with significant fibrosis (F ≥ 2 on the METAVIR scale) established by transient elastography (TE) at the start of the treatment, and we compared them with patients without liver fibrosis (F0/F1) at baseline. 

### 2.2. Transient Elastography (TE)

TE was performed by trained examiners, certified by the manufacturer, using the FibroScan device (Echosens, Paris, France). Two different probes were used: medium (M) and XL (for obese patients). TE was performed in patients who had fasted for at least 2 h. The liver stiffness measurement (LSM) and controlled attenuation parameter (CAP) were simultaneously obtained. The adequacy of the measurement was assessed by the FibroScan device (Echosens, Paris, France). The examination was considered successful when 10 valid measurements were conducted with at least a 60% success rate and an interquartile range (IQR) of less than 30% of the median LSM value [12]. The final LSM result was expressed as the median value of at least 10 valid measurements, and it was assessed in kilopascals (kPa). It corresponded to liver fibrosis on the METAVIR scale according to the Castera TE cutoffs as follows: no to mild fibrosis (F0-F1), LSM up to 7.0 kPa; moderate fibrosis (F2), LSM 7.1 to 9.4 kPa; severe fibrosis (F3), LSM 9.5 to 12.4 kPa; and cirrhosis (F4), LSM ≥ 12.5 kPa [13]. Liver fibrosis was considered significant if the LSM median was >7 kPa, corresponding to a METAVIR F score ≥ 2 points. The final CAP values ranged between 100 and 400 decibels per meter (dB/m) and were assigned as follows: no steatosis (S0, CAP 0–238 dB/m), mild steatosis (S1, CAP 239–260 dB/m), moderate steatosis (S2, CAP 261–290 dB/m), and severe steatosis (S3, CAP > 290 dB/m) [10,14]. In patients with significant fibrosis at baseline, TE examination was performed three times: on the day the patient started treatment, at week 12 posttreatment, and one year after the end of the treatment, simultaneously with the biomarker evaluation. In patients without liver fibrosis at the start of the treatment, TE assessments were not routinely performed.

### 2.3. Biomarker Evaluation

Biochemical serum testing was performed using commercially available laboratory kits. For both alanine and aspartate aminotransferase (ALT and AST) serum levels, 40 IU/L was considered the upper limit of normal (ULN). Two indirect fibrosis biomarkers were calculated, namely, the aspartate transaminase-to-platelet ratio index (APRI, Equation (1)) and fibrosis-4 index (FIB-4, Equation (2)), according to the published analytic recommendations [15,16]: (1)APRI=[(AST (IU/L)/AST ULN)/Platelet count (10^9/L)]×100
(2)FIB-4=[Age (years)×AST (IU/L)]/[Platelet count (10^9/L)×ALT (IU/L)]

According to previously published data, the following cutoffs for the biomarkers were considered: APRI > 0.5 and FIB-4 > 1.45, which suggests significant fibrosis, and APRI > 1.5, which suggests cirrhosis [15,17].

### 2.4. Ethical Statement

Written informed consent was collected from all the patients and/or their parents/guardians before their inclusion in the study. The investigation was performed in accordance with the ethical standards in the 1964 Declaration of Helsinki and its later amendments. The local ethics committee of the Medical University of Warsaw approved this study (No KB/87/2019; date of approval: 13 May 2019).

## 3. Results

### 3.1. Participants

Among the 35 patients included in this therapeutic program, there were four patients aged 12–17 years with significant fibrosis: three with cirrhosis (F4 on the METAVIR scale), including a 12-year-old girl (Patient 1), 16- and 17-year-old boys (Patients 2 and 3); and one 15-year-old male patient presenting with a fibrosis score of 2 on the METAVIR scale (Patient 4). Their Child–Pugh class was A for all three cirrhotic patients. All four patients were infected vertically from an HCV-infected mother. Three subjects were infected with genotype 1b of the HCV and one with genotype 4. Patient 2 was coinfected with human immunodeficiency virus (HIV), and he was effectively treated with tenofovir alafenamide/emtricitabine/raltegravir (TAF/FTC/RAL). All patients were treated with a fixed dose of 90 mg/200 mg LDV/SOF. Two patients were qualified for a 12-week treatment, and two for a 24-week therapy. The baseline characteristics of these patients are presented in Table 1. We compared them with a group of four patients without liver fibrosis at baseline, in which liver fibrosis was assessed with TE one year after the end of the treatment. They were similar in age to the previous group, and one patient was also coinfected with HIV (treated effectively with TAF/FTC/rilpivirine (RPV). Baseline medians of the LSM in these four patients were between 4.6 and 6.1 kPa, CAP between 154 and 212 dB/m, APRI 0.273 to 0.419, and FIB-4 0.204 to 0.422. Baseline characteristics of this group are presented in Table 1.

### 3.2. Treatment Outcomes

Four weeks after the start of treatment, the HCV viral load was undetectable in all eight patients. A decrease in the ALT level was observed in all patients, except one child with an initial ALT level of 18 IU/L; in her case, it remained normal (29 IU/L). At the end of the treatment, five patients had an undetectable HCV viral load (including two with significant fibrosis), and it was unavailable in the remaining three subjects because their visits had to be canceled due to the coronavirus disease (COVID-19) pandemic. A similar situation was observed at 12 weeks posttreatment (SVR assessment). One year after the end of treatment, the viral load was undetectable in seven patients (including three with significant fibrosis), which confirmed the efficacy of the treatment. ALT levels in all these patients were normal. One patient with cirrhosis at baseline (Patient 3) did not show up for the following visits after week 4 and was thus lost to follow-up. 

The LDV/SOF therapy was well tolerated. No serious adverse events were observed. During the first 2 to 6 weeks of treatment, two patients (in the group with significant fibrosis) reported headache, two reported fatigue, and one reported diarrhea, which resolved spontaneously. 

### 3.3. Liver Fibrosis and Steatosis after the Treatment

Observations performed at week 12 posttreatment in the group of patients with significant fibrosis at baseline revealed a significant improvement in the LSM in Patient 1, corresponding to a decrease from the F4 to the F0-F1 stage on the METAVIR scale (Figure 1). In Patient 2, the increase in the LSM was from 14 to 33.6 kPa (F4). However, in both these patients, a decrease in the APRI and FIB-4 values was observed (Figure 1). In Patient 1, the LSM improvement was accompanied by a decrease in the CAP from S1 to S0, whereas in Patient 2, an increase in the CAP was found (Figure 1). Due to the COVID-19 pandemic, the 12-week posttreatment visits for Patients 3 and 4 had to be canceled. Thus, data for these two patients were unavailable. 

At the final visit, one year after completing the treatment, the LSM medians corresponded to F0-F1 in Patients 1 and 4, whereas in Patient 2, the LSM was lower compared to the previous visit but still corresponded to an F4 score on the METAVIR scale. However, the APRI values decreased for all three of these patients (Figure 1). The CAP value for Patient 1 increased compared to the 12-week posttreatment visit, but it was lower compared to the initial visit. A decrease in the CAP was observed for Patient 2 (from stage S3 to stage S1). However, in both of these patients, their BMI z-scores were over 1.0, suggesting obesity (Patient 1) or overweight (Patient 2). In Patient 4, the CAP correlated to no steatosis (Figure 1). Patient 3 did not show up for this visit and was consequently lost to follow-up. 

Among the patients without liver fibrosis at baseline, TE evaluation one year after the end of treatment did not reveal any significant changes; the medians of the LSMs were between 4.1 and 6.0 kPa, corresponding to F0/F1 on the METAVIR scale. The CAP was below 238 dB/m in three patients, and it corresponded to steatosis stage S2 (275 dB/m) in one patient, which may be due to weight gain (his BMI increased from 23.5 to 31.2 kg/m^2^). In these four patients, the APRI and FIB-4 values one year after the end of the treatment were similar to the baseline (0.180 to 0.303, and 0.312 to 0.647, respectively).

## 4. Discussion

A fixed-dose combination of LDV/SOF is approved by the European Medicines Agency and Food and Drug Administration for the treatment of chronic HCV infection with genotypes 1, 4, 5, and 6 in children and adolescents 3 years of age and older [18,19,20]. Its safety and efficacy have been confirmed in phase II and III clinical studies [18,19,20]. However, there are only limited data available on the LDV/SOF efficacy in children with significant fibrosis [9]. In addition, little is known about the long-term outcomes and the influence of the antiviral treatment on liver fibrosis. Patients with confirmed liver fibrosis should be closely monitored even after effective antiviral treatment of CHC [21]. Interestingly, there is evidence that in adult patients, liver fibrosis may be to some extent reversed by DAA treatment [21,22,23,24]. Bachofner et al. revealed a 32% reduction in the LSM after DAA treatment in 392 adult patients with CHC complicated by fibrosis [25]. However, patients with significant fibrosis are still at risk of HCC development even after achieving SVR [21,26]. Mogahed et al. recently reported an improvement in the LSM in pediatric patients with CHC after treatment with DAA [10]. They studied 23 Egyptian children infected with HCV genotype 4 aged 10 to 18 years with variable degrees of fibrosis at baseline and found a significant improvement in the LSM, APRI, and FIB-4 one year after SVR achievement. In 13 (56.5%) of their patients, the LSM improved; in seven patients, it was stationary; and the remaining three subjects showed a mild increase in the LSM, with improvement in the APRI and FIB-4 [10]. This observation is consistent with the results of our study. In two out of four patients with significant liver fibrosis at baseline, an improvement and regression of liver fibrosis were revealed (from stage F4 and F2 to F0-F1 on the METAVIR scale) one year after the end of LDV/SOF therapy. In one patient, the LSM median increased 12 weeks after the end of the treatment and then decreased, but it still correlated to stage F4. In all these patients, an improvement in the APRI was observed. Similar observations were reported by Makhlouf et al.; in their work, 65 adolescents with CHC genotype 4 were treated with LDV/SOF for 12 weeks [27]. At SVR12, they observed a significant improvement in liver stiffness, measured by shear wave elastography, and the APRI. In 14 out of 65 (21.6%) patients, there was a transition in the stage of fibrosis observed: in 10 cases from F1 to F0, in three cases from F2 to F1, and in one case from F3 to F2 [27]. In our patients without liver fibrosis at baseline, no progression of liver stiffness was observed one year after the end of treatment. Their LSM evaluations corresponded to the F0-F1 stage on the METAVIR scale, whereas their APRI and FIB-4 remained below the cut-offs for significant fibrosis. Thus, one year after the end of successful treatment with LDV/SOF, no fibrosis (F0-F1) was observed in six of our patients, including four subjects presenting without liver fibrosis at the start of the treatment, and two with significant fibrosis at baseline (Figure 2). In one patient with cirrhosis at baseline, no improvement was observed, and one subject was lost to follow-up (Figure 2).

Factors that could influence the regression of liver fibrosis after successful DAA treatment remain unknown. In the study by Mogahed et al., comorbidities or previous ineffective treatment with interferon were not associated with an increased LSM one year after SVR had been achieved [10]. Among our patients, the one subject in which the LSM increased after completing the LDV/SOF treatment was coinfected with HIV, but he received effective antiretroviral treatment with TAF/FTC/RAL, and he had no immunodeficiency. He had been previously ineffectively treated with interferon with ribavirin and was older compared to patients who achieved regression of fibrosis. However, available data on regression of liver fibrosis after curing CHC with DAAs in adult patients with and without HIV coinfection, suggest that the dynamics of liver fibrosis regression are not influenced by HIV coinfection [28]. Thus, other than HIV infection, contributors to liver fibrosis need to be considered.

In our recent study, we demonstrated a good correlation between TE and biomarkers for the staging of liver fibrosis in pediatric patients with CHC, but we suggested that the fibrosis cutoffs for the APRI and FIB-4 might be lower for adolescents than for adults [6]. Thus, the APRI and FIB-4 values in our cirrhotic patients were lower than the usually considered cutoffs [15,17]. The APRI decrease correlated with the regression of liver fibrosis evaluated by TE in two of our patients, but a discordance was found for Patient 2. In his case, an increase in the LSM was accompanied by a decrease in the APRI, which suggests that further observations are necessary to validate the usefulness of biomarker evaluation for monitoring of liver fibrosis after DAA treatment. 

The COVID-19 pandemic, which was announced in March 2020, has led to the disruption of the healthcare system and has had a significant negative impact on the care of patients with chronic diseases, including our therapeutic program for children with CHC. In March 2020, our department was transformed to a setting dedicated to COVID-19 patients. Consequently, all visits for non-COVID-19 patients between March and July 2020 had to be canceled or postponed. However, we have prepared and followed the new guidelines for management of children and adolescents with CHC during the COVID-19 pandemic [29]. Several efforts were made to prioritize patient care in our children with CHC. In the case of patients receiving DAA therapy, these efforts included using telemedicine for monitoring the patient’s general condition, adherence to treatment, and the side effects of the therapy, at least every four weeks; cooperating with general practitioners; using local laboratory testing for follow-up testing; and engaging in the home delivery of DAAs [29]. All these efforts resulted in completing the full treatment regimen by all our patients. However, several monitoring visits were unfortunately canceled; this problem in particular appeared in Patient 3, who was the last patient included in the therapeutic program (in February 2020). He completed his visit at four weeks of treatment, but the next monitoring visits were canceled and replaced by phone calls every four weeks. Home delivery of LDV/SOF was arranged for him until the end of treatment. After 12 weeks of treatment, his ALT and AST were tested in the local laboratory, which revealed further improvement in their levels (120 and 44 IU/L, compared to 438 and 184 IU/L at the start of treatment, and 272 and 111 IU/L after four weeks). The patient was asked to attend follow-up visits to assess the SVR and long-term outcome of treatment, but due to family and social problems and the long distance from our center, he refused to come and was lost to follow-up. 

The main limitation of this study was the low number of patients included in the final analysis. However, to the best of our knowledge, there is only one report available on the long-term effects of DAA treatment in children including the influence of the therapy on liver fibrosis [10]. Therefore, it is worthwhile to share our unique data. In most pediatric cohorts, cirrhotic patients are underrepresented; thus, finding larger groups of these specific patients would be difficult. In addition, there were only four patients without fibrosis included in this study due to the unavailability of data for other patients, as in subjects without liver fibrosis at the start of the treatment, TE assessments had not been routinely performed. However, it is worthwhile to underline that both groups of patients in this study (presenting with and without liver fibrosis at baseline) matched with respect to age and HIV coinfection. The second issue is the gaps in the available data due to the disruption caused by the COVID-19 pandemic. However, it is worth noting that DAA therapies are simple, short, and safe. Thus, even when close monitoring of patients is not possible, positive outcomes of treatment may be achieved. 

Based on our experience, we conclude that treatment with a fixed-dose LDV/SOF in children with significant fibrosis in the course of CHC is safe and effective. A one-year observation of these patients after the end of treatment revealed that regression of liver fibrosis, including cirrhosis, is possible but not certain. Further observations in larger groups of patients are necessary to find predictors of liver fibrosis regression in pediatric patients with CHC.

## Figures and Tables

**Figure 1 viruses-13-01518-f001:**
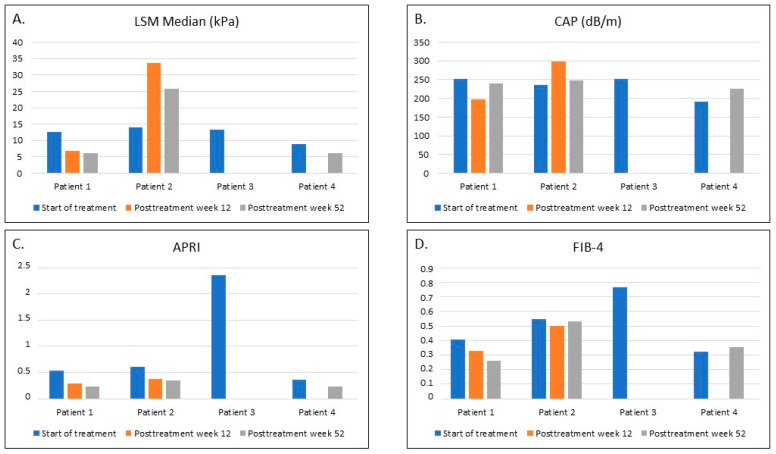
Noninvasive evaluation of liver fibrosis and steatosis in four patients with significant fibrosis at baseline, treated with ledipasvir/sofosbuvir: at the start of treatment, at week 12, and week 52 posttreatment. (**A**). Liver stiffness measurement (LSM) median. (**B**). Controlled attenuation parameter (CAP) (**C**). Aspartate transaminase-to-platelet ratio index (APRI) (**D**). Fibrosis-4 index (FIB-4).

**Figure 2 viruses-13-01518-f002:**
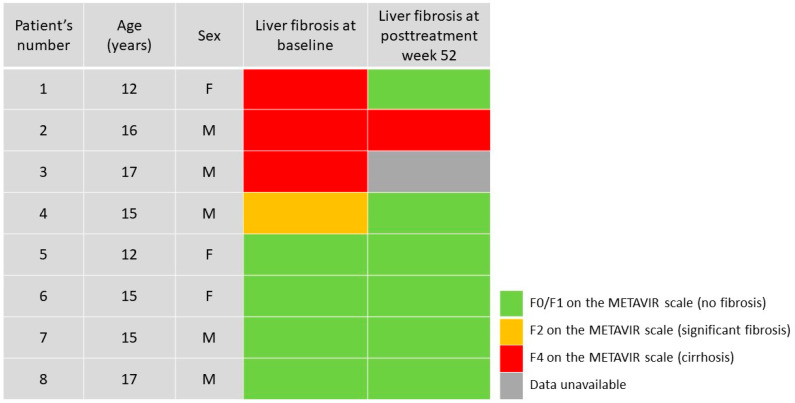
Stages of liver fibrosis evaluated by transient elastography at baseline and one-year after the end of treatment with ledipasvir/sofosbuvir in 8 patients. Each row represents one patient. F—female; M—male.

**Table 1 viruses-13-01518-t001:** Clinical and laboratory characteristics of patients with chronic hepatitis C treated with ledipasvir/sofosbuvir according to their baseline stage of fibrosis.

Feature		Patients with Significant Fibrosis (F ≥ 2 on METAVIR Scale) *n* = 4	Patients without Fibrosis (F0/F1 on METAVIR Scale) *n* = 4
Sex	Male	3	2
	Female	1	2
Age at start of the treatment (years)		12; 16; 17; 15	12; 15; 15; 17
HCV genotype	1	3	3
	4	1	1
Mode of HCV infection	Vertical	4	3
	Unknown	0	1
Previous ineffective anti-HCV treatment (interferon plus ribavirin)		3	2
Duration of LDV/SOF treatment	12 weeks	2	4
	24 weeks	2	0
BMI (kg/m^2^)/BMI z-score at start of LDV/SOF		25.4/2.07; 25.7/1.55; 37.0/3.28; 20.4/0.23	18.0/0.01; 20.9/0.23; 18.4/−0.64; 23.5/0.77
ALT (IU/mL) at start of LDV/SOF		40; 52; 438; 46	32; 41; 18; 67
HCV/HIV coinfection		1	1
HCV viral load (IU/mL) at start of LDV/SOF		2.23 × 10^6^; 4.89 × 10^5^; 7.0 ×10^4^; 6.28 × 10^5^	4.49 × 10^5^; 1.37 × 10^4^; 7.06 × 10^5^; 2.24 × 10^6^
Undetectable HCV viral load ≥ 12 weeks posttreatment (SVR)		3 (one patient lost to follow-up after week 4)	4

Data are presented as numbers of patients, unless otherwise indicated. ALT—alanine aminotransferase; BMI—body mass index; HIV—human immunodeficiency virus; LDV/SOF—ledipasvir/sofosbuvir; SVR—sustained virologic response.

## Data Availability

The datasets used and analyzed during the current study are available from the corresponding author upon reasonable request.

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
