# Peer review of "One-Year Outcomes after Ledipasvir/Sofosbuvir Treatment of Chronic Hepatitis C in Teenagers with and without Significant Liver Fibrosis—A Case Series Report"

_viruses, 2021, doi:10.3390/v13081518_

Round 1

Reviewer 1 Report

In their manuscript, Pokorska-Ĺšpiewak et al describe the evolution of non-invasive markers of liver fibrosis in 4 children with ≥F2 fibrosis prior to initiating DAA with ledipasvir/sofosbuvir. For the three patients with available measurements, two had declines in all three markers (i.e. APRI, FIB-4, TE) while one had an increase in TE but not APRI or FIB-4. The manuscript reads nicely as a case series.

One major issue is that there are so few patients. The authors contend that data of this kind are rare (which is indeed the case). But why only include individuals with significant fibrosis or higher? and not all 35 children qualifying for treatment? This will give clinicians a more comprehensive perspective as to what happens to these children across the spectrum of liver disease. Furthermore, the increase in liver fibrosis of the one patient begs the question whether it can also be observed in patients with liver fibrosis levels at F0-F1.

There is little insight as to why the one patient even observed an increase in only TE. The authors highlight HIV co-infection, but what aspects of HIV-infection would cause a high TE measurement? What agents were this child exposed to (i.e. D-drugs? Indinavir? Atazanavir?)? Was the nadir CD4 cell counts low? Any AIDS-defining illnesses? The authors should look into cohort studies that have examined fibrosis regression/progression in HIV-HCV co-infected individuals and expand their discussion.

Also, why was there such a discordance with the TE measurements versus the other non-invasive markers of liver fibrosis? This was never fully reconciled in the text.

Finally, the authors also repeatedly mention that these data are long-term. 52 weeks after end of treatment is not long term. Please reword.

Minor comments:

- ln 20. It needs to be stated here that one person was lost to follow-up 4 weeks after EOT.

- ln 39. The increase of 0.09 kPa is not at all clinically relevant. Does this mean that all the patients included in this study remained at the same level of fibrosis? And over what period of time?

- ln 116. According to WHO guidance, the FIB-4 has two cutoffs for F2 fibrosis: ≥1.45 and ≥3.25. Do the authors obtain the same results with the alternative cutoff?

- ln 131. Was the HIV-positive child on antiretroviral medication? If so, which agents?

- Table 1. What is the difference between “<12” and “undetectable”? “<12” means that the target was detected but levels were <12? For sake of clarity, undetectable HCV RNA should be defined by <12 IU/mL.

- Figure 1. There is a fair amount of overlap with Table 1. Suggest parsing Table 1.

Reviewer 2 Report

In this manuscript, Pokorska-Ĺšpiewak and collaborators analyzed the outcome of the ledipasvir/sofosbuvir therapy on 4 adolescent HCV patients with severe fibrosis. Although the topic is relevant and offers interesting hints, in my opinion the limited number of patients and the different results obtained in the 4 patients render this study of scarce clinical relevance. 

The authors should find the way of increasing the sample size, or consider also patients with different ages to corroborate their findings. therefore, the paper is not acceptable in the present form. 

Round 2

Reviewer 1 Report

I thank the authors for the revised manuscript and their responses to my comments. I still have one major concern, which can be easily resolved. 

It was a good idea to include a comparator group without liver fibrosis, but how exactly was this group selected? In table 1, it states that there were only 4 of these individuals from the supposedly 31 children without liver fibrosis. Was there matching involved? Or was this based on data availability? This needs to be clearer, as this group might not represent the overall population of children without cirrhosis. Also, the comparison between 4 and 4 individuals is highly underpowered and is purely descriptive. These aspects need to be added as limitations.

Other minor comments: 

- ln 111-112. Should be: "In patients without liver fibrosis at the start of the treatment, TE assessments were not routinely performed."

- ln 139-141. Was TAF/FTC/RAL the only ART regimen this patient had ever received? The same question applies to the individual described in ln 147 with TAF/FTC/RPV. It depends on the route of HIV tranmission, but if these children were vertically infected at birth, I highly doubt that this regimen would have been their first. Also, was previous exposure to D-drugs involved? Again, other contributors to liver fibrosis NEED to be considered.

- ln 176. It needs to be clear that the first part of this section only refers to the children with liver fibrosis at baseline.

- Figure 2. It would make more sense to transpose the columns and rows. Also, a column of the transposed figure should be added with patient information (i.e. those with and without liver fibrosis). Ideally, patient numbers should correspond to the other figures.

Reviewer 2 Report

The manuscript has been sufficiently improved for publication. In particular, the definition "case series report" present in the revised version of the title is in line with the limited number of patients.
